# Proper Backward Connection Placement Boosts Spiking Neural Networks

## Abstract

We study how backward connections (BCs, also known as temporal feedback connections) impact the performance of Spiking Neural Networks (SNNs) and how to effectively search the placement of BC to boost SNNs' performance. Presumably, BCs have the potential to enhance SNNs' representation capacity by creating new temporal pathways in the SNN. However, we empirically find that BCs at different positions are not equally beneficial to SNNs, and some of them may even lead to performance degradation. Given the large search space of BCs placement, we propose Backward Connection Neural Architecture Search (BCNAS-SNN), a framework that automatically identifies the optimal BCs pattern by searching all possible BCs including both intra-block and inter-block ones. Extensive experiments indicate that BCNAS-SNN is able to achieve state-of-the-art results on the CIFAR10, CIFAR100, and Tiny-ImageNet datasets, with accuracy of 95.67%, 78.59%, and 63.43%, respectively. A set of ablation studies are further presented to understand the efficacy of each design component in BCNAS-SNN.

## 1 Introduction

Spiking neural networks (SNNs) are considered the third generation of neural networks Maass (1997); Roy et al. (2019); Christensen et al. (2022) due to their unique properties, such as asynchronous computation, low power consumption Akopyan et al. (2015); Davies et al. (2018), and inherent temporal dynamics. SNNs transmit information in the form of binary spikes during multiple time steps and thus enjoy the advantage of multiplication-free inference against artificial neural networks (ANNs).

To improve the performance of SNNs, many efforts have been made to design various training algorithms and spiking neurons. Researchers have focused on the surrogate gradient for the firing function Wu et al. (2018); Bellec et al., the loss function Deng et al. (2021), and the normalization layers Zheng et al. (2021); Duan et al.. However, relatively few studies have explored the architecture design of SNNs, particularly with regard to backward connections (BCs) Panda et al. (2020). SNNs are composed of Leaky-Integrate-and-Fire (LIF) Izhikevich (2003) neurons, which store and transmit temporal information. The backward connection is a promising component to make better use of this temporal information in SNNs. For instance, SRNNs Yin et al. (2020), in which each layer has recurrent connections, is able to achieve better performance than its counterpart without BC. BackEISNN Zhao et al. (2022) applies adaptive time-delayed self-feedback to regulate the precision of spikes.

However, these studies use fixed backward connections for each spiking neuron. SNASNet Kim et al. (2022) leverages a cell-based Neural Architecture Search (NAS) method to search the inter-block BCs. While the question of how to effectively add global backward connections to SNNs remains an open problem.

In this paper, we aim to study the global backward connections in SNNs which can link any two layers regardless of feature resolution. To enable such a global BC, we leverage an upsample layer and a $1 \times 1$ spiking convolutional layer to send the activations (Figure 2.A). In order to locate the optimal global BCs, we propose BCNAS-SNN, an efficient BCs searching framework based on an evolutionary algorithm. Based on the characteristic of BCs, we introduce two additional design components to improve and accelerate the searching process: 1) We carefully craft the search space

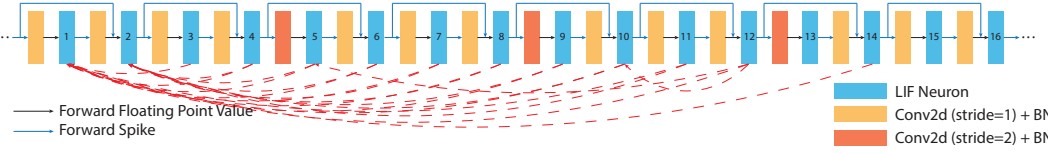

Figure 1: **The optimal BCs are always connected to the front LIF layers.** We have drawn all searched backward connections based on ResNets across various datasets, including CIFAR10/100, CIFAR10DVS and Tiny-Imagenet. More details are included in Appendix C.

and the mutation/crossover for BCs to balance exploration and exploitation; 2) We initialize the evolutionary algorithm by score-based selection from a large population of random architectures.

Extensive experiments on a variety of datasets demonstrate that BCNAS-SNN is able to discover appropriate BCs, which improve standard SNNs' accuracy. Additionally, as shown in Figure 1, an intriguing phenomenon is that the searched backward connections prefer to be connected to the beginning of SNNs which motivates future architecture design of SNNs. In summary, our key contributions are:

- To the best of our knowledge, this work is the first study thoroughly investigating the inequal effects of global backward connections in SNNs.

- We develop an efficient spike-based NAS framework with well-crafted search space and strategy. The score-based selective initialization allows the evolutionary search algorithm to converge fast and obtain optimal BCs.

- Extensive experiments on both static datasets and neuromorphic datasets demonstrate the importance of appropriate BCs for SNNs and validate the effectiveness of our proposed search framework. For the CIFAR10/100 and Tiny-ImageNet datasets, the searched architectures achieve top-1 accuracy of 95.67%, 78.59%, and 63.43% for the first time, outperforming existing state-of-the-art results.

## 2 RELATED WORK

### 2.1 SPIKING NEURAL NETWORKS

The binary spike mechanism in SNNs leads to the non-differentiability of the activation function Neftci et al. (2019), which disables the typically used gradient descent approaches. Direct training approaches utilizing surrogate gradients Wu et al. (2018); Shrestha & Orchard (2018); Fang et al. (2021); Li et al. (2021); Deng et al. (2021) have demonstrated superior performance with smaller time steps. For instance, the RecDis-SNN method proposed in Guo et al. (2022) rectifies the membrane potential distribution to enhance performance. Additionally, the IM-Loss method presented in Guo et al. maximizes the information flow in order to enhance expressiveness. Furthermore, the GLIF method Yao et al. introduces a gate mechanism to increase the representation space of spiking neurons.

In addition to designing training algorithms and spiking neurons, some studies focus on the architecture of SNNs. NASSNN Kim et al. (2022) and SpikeDHS Che et al. adopt NAS to search optimal SNN architectures based on ANN search spaces Liu et al. (2018); Dong & Yang (2019). AutoSNN Na et al. (2022) utilizes NAS to search different convolution blocks in SNNs. In this study, we will illustrate that properly incorporating one or two backward connections to ResNet He et al. (2016) structures will improve SNNs significantly.

### 2.2 NEURAL ARCHITECTURE SEARCH

Neural Architecture Search (NAS) was proposed to automatically select high-performance neural networks. At the early stage, NAS methods Baker et al. (2016); Zoph & Le (2016); Zoph et al. (2018) needed to train candidate architectures from scratch which took much time. To alleviate the expensive

search cost, many lightweight NAS methods have been proposed, including one-shot weight-sharing NAS and zero-cost (ZC) proxies.

For one-shot weight-sharing NAS Pham et al. (2018); Guo et al. (2020); Cai et al. (2020); Yan et al. (2021), the search process can be divided into two phases: training the super-network and evaluating the candidate architectures. In the training phase, architectures are randomly sampled in each training loop, and all candidate architectures in the search space can be roughly trained. In the evaluation phase, architectures inheriting weights from the trained super-network can be directly evaluated on the validation dataset. To further improve the search efficiency, ZC proxies were introduced to evaluate all candidate architectures without training Krishnakumar et al.. Mellor et al. (2021) estimated the separability of the minibatch into different linear regions of the output space. Abdelfattah et al. (2020) proposed ZC proxies inspired by pruning-at-initialization techniques Lee et al. (2018); Turner et al. (2019); Tanaka et al. (2020); Wang et al. (2019). Some works Chen et al. (2020); Xu et al. (2021); Zhu et al. were based on neural tangent kernels Arora et al. (2019); Du et al. (2019). Shu et al. (2022) presented a unified theoretical analysis of gradient-based training-free NAS and developed a hybrid NAS framework.

## 3 PRELIMINARIES

### 3.1 SPIKING NEURON MODEL

We adopt the iterative LIF model Wu et al. (2019); Deng et al. (2021) and the membrane potential is updated as:

$$\mathbf{v}(t+1) = \tau \mathbf{v}(t) + \mathbf{I}(t), \tag{1}$$

where $\mathbf{v}(t)$ denotes the membrane potential at time step $t$, $\tau$ is the leaky factor, and $\mathbf{I}(t)$ represents the pre-synaptic inputs, which is the product of synaptic weight $\mathbf{W}$ and spiking input $\mathbf{x}(t)$. Given the threshold $V_{th}$, the neuron fires a spike and $\mathbf{v}(t)$ resets to 0 when it exceeds $V_{th}$. The spike firing and hard reset mechanism can be described as:

$$
\begin{aligned}
\mathbf{s}(t+1) &= \mathbf{\Theta}(\mathbf{v}(t+1) - V_{th}) \\
\mathbf{v}(t+1) &= \mathbf{v}(t+1) \cdot (1 - \mathbf{s}(t+1)),
\end{aligned}
\tag{2}
$$

where $\mathbf{\Theta}(\cdot)$ is the Heaviside step function. The output spike $\mathbf{s}(t+1)$ will become the post-synaptic spike and propagate to the next layer. In this work, we set $V_{th}$ to 1 and $\tau$ to 0.5 for all experiments.

### 3.2 LOSS FUNCTION

Many previous works define the loss function in SNNs by cross entropy $\mathcal{L}_{CE}$ Zheng et al. (2021) or mean squared error $\mathcal{L}_{MSE}$ Fang et al. (2021). Deng et al. (2021) propose a new loss function, called TET loss, making SNNs converge quickly, which is described as:

$$\mathcal{L}_{TET} = \frac{1}{T} \sum_{t=1}^{T} \mathcal{L}_{CE}[\mathbf{O}(t), \mathbf{y}], \tag{3}$$

where $\mathbf{O}(t)$ represents the pre-synaptic input of the output layer and $\mathbf{y}$ is the target label in classification tasks. In this work, we use TET loss for all experiments.

### 3.3 SURROGATE GRADIENT FUNCTION

By applying the chain rule, we backpropagate the gradients of the loss $\mathcal{L}$ on both spatial and temporal domains Wu et al. (2018) as follow:

$$\frac{\partial \mathcal{L}}{\partial \mathbf{W}} = \sum_t \frac{\partial \mathcal{L}}{\partial \mathbf{s}(t)} \frac{\partial \mathbf{s}(t)}{\partial \mathbf{v}(t)} \frac{\partial \mathbf{v}(t)}{\partial \mathbf{I}(t)} \frac{\partial \mathbf{I}(t)}{\partial \mathbf{W}}. \tag{4}$$

However, the gradient of the spiking function, represented by $\frac{\partial \mathbf{s}(t)}{\partial \mathbf{v}(t)}$, does not exist. To circumvent this obstacle, a technique known as the Surrogate Gradient (SG) approach is employed. This technique approximates the true gradients through the utilization of continuous functions. In this particular

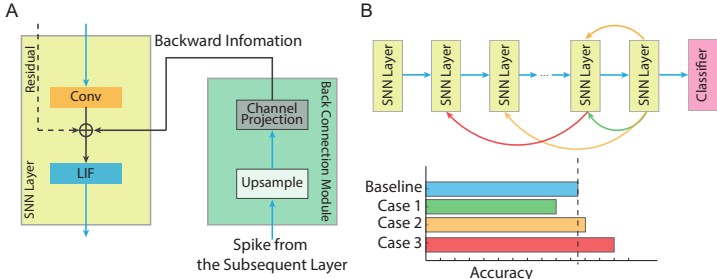

Figure 2: **Concept of backward connections and results under randomly adding backward connections.** The backward module contains an unsample layer and a spiking convolutional layer to project the backward spike tensors. Then, we randomly select three architectures with 1 or 2 backward connections and compare their test accuracy against the hand crafted one without backward connections. This indicates the preference of searching BCs against random adding or hand crafting them.

study, we utilize the triangle curve Bellec et al. (2018) as the surrogate gradient, which can be described as:

$$\frac{\partial \mathbf{s}(t)}{\partial \mathbf{v}(t)} = \frac{1}{b^2} max(0, b - |\mathbf{v}(t) - V_{th}|), \tag{5}$$

where $b$ determines the triangle sharpness and is set to 1.

## 4 METHODOLOGY

### 4.1 GLOBAL BACKWARD CONNECTIONS

In this section, we introduce our definition of global backward connection in SNNs. Prior arts have been using backward connections for the same LIF layer or LIF layers with the same resolution Panda & Roy (2017); Demin & Nekhaev (2018); Zhang & Li (2019); Panda et al. (2020); Yin et al. (2020); Zhao et al. (2022); Kim et al. (2022) and usually the network is shallow. This backward connection, however, is essentially a "recurrent" connection that is restricted to transmitting the information in a local manner. Moreover, it cannot be applied to the previous layers that share different feature resolutions. Therefore, we argue that the potential for backward connections has not been fully exploited.

In this study, we take ResNet as the network backbone and treat each LIF layer as a port. A global backward connection can exist between any two ports, including the case where the two ports are the same. The ports in the back layer and the front layer are regarded as the output port and input port of the backward connection, respectively. We adopt fixed operations on each global backward connection (Figure 2.A), including nearest interpolation, $1 \times 1$ convolution, and batch normalization, which can deal with two LIF layers with different feature resolutions.

Formally, the global backward connection from any layer $j$ to layer $i (j \geq i)$ will charge the membrane potential in the layer $i$ as well, given by:

$$\mathbf{I}^{(i)}(t) = \mathbf{W}^{(i)}\mathbf{s}^{(i-1)}(t) + \mathbf{W}^{(*)}\gamma(\mathbf{s}^{(j)}(t-1)), \tag{6}$$

where $\mathbf{W}^{(i)}$ represents the forward weight connection between layer $i - 1$ and layer $i$, $\mathbf{W}^{(*)}$ is the backward weight connection, $\mathbf{s}^{(i-1)}(t)$ denotes the output spike of layer $i - 1$ at time step $t$, and $\gamma(\cdot)$ indicates the nearest interpolation operation. By using this global backward connection, we can largely extend the possible network topology in the temporal dimension.

However, we generally discern that not all global backward connections can improve the SNN performance. To demonstrate that, we randomly sample 1 or 2 backward connections to a ResNet-18 on the CIFAR-100 dataset. Figure 2.B shows 3 examples as well as their final accuracy. It can be

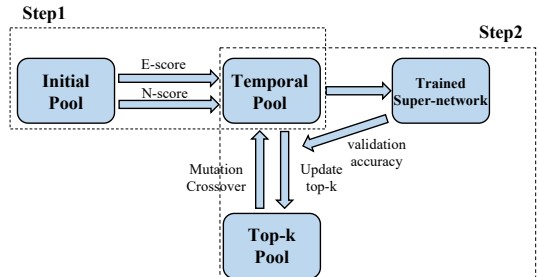

Figure 3: **Two-step search strategy of BCNAS-SNN.**

found that backward connections may lead to an accuracy decrease more than 6%, which means finding good backward connections that maximize the accuracy of SNN is non-trivial.

## 4.2 BCNAS-SNN

To discover BCs that greatly improve the network performance of SNNs, we develop a search framework based on an evolutionary algorithm, called BCNAS-SNN. Inspired by the one-shot weight-sharing method in NAS Guo et al. (2020); Na et al. (2022), we use the validation accuracy as the ultimate metric for each candidate architecture. According to the similarity between different BCs, we define the evolutionary mechanism, including mutation and crossover, to efficiently search optimal BCs in potential. Meanwhile, the initial population is crucial to the results of evolutionary algorithms, which need an efficient approach to obtain. Many previous works directly use random samples and some of which adjust their population size to be very large leading to the expensive time consumption in the search process. By contrast, we propose the score-based method that quickly collects a small number of superior BCs as the initialization, which is proved to be effective in the experiments.

We take the two-step search strategy for BCNAS-SNN as shown in Figure 3. First, we simultaneously leverage two different training-free scores choosing some BCs prepared for the next step. Then, we adopt an accuracy-based evolutionary algorithm to further search optimal BCs. The rest content is divided into three parts. In subsubsection 4.2.1, we define the search space for BCNAS-SNN. The score-based method and the accuracy-based evolutionary algorithm are detailed in subsubsection 4.2.2 and subsubsection 4.2.3, respectively.

### 4.2.1 SEARCH SPACE

The design of a search space is crucial in NAS and can significantly affect the search results. In this study, we use ResNet as a representative network backbone. ResNet has a fixed number of LIF layers and thus the number of all possible backward connections is determined. To minimize the complexity and computational cost of searching, we fix the operations on each backward connection and set the maximum number of backward connections for each SNN to $b$. When the number of LIF layers is $m$, we can obtain $n = \frac{m(m+1)}{2}$ possible backward connections in total. The size of the search space $\mathcal{A}$ can be calculated as $|\mathcal{A}| = \sum_{i=0}^{b} \binom{n}{i}$. In practice, we find that setting N as 2 is sufficient to boost the performance of SNNs.

### 4.2.2 SCORE-BASED SELECTIVE INITIALIZATION

**N-score & E-score.** In order to efficiently find a good initial population for the evolutionary algorithm, we propose the training-free score-based method inspired by zero-shot methods Abdelfattah et al. (2020); Montufar et al. (2014); Lopes et al. (2021). Recently, Krishnakumar et al. prove that different zero-shot methods do indeed compute substantial complementary information. Thus, we consider adopting a combined method to initialize the population. NASWOT Mellor et al. (2021) observed that the initialized network having distinctive representations across different data samples is likely to achieve higher performance and Kim et al. (2022) succeeded in applying it to SNNs. We utilize this approach to calculate N-score in this work. However, the computation of the N-score ignores the

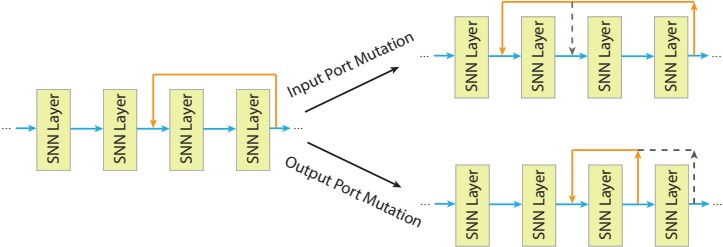

Figure 4: **Mutation in the evolutionary algorithm.** For simplicity, we use a 4-neuron LIF network as an example. In our evolutionary algorithm, backward connections may result in both output port mutation and input port mutation.

gradient information of the initialized network, which is important for an accurate evaluation of the architecture. And we need to leverage another zero-shot method to gain a score as the supplement of gradient information. EPE-NAS Lopes et al. (2021) evaluated how the gradients behave with respect to the input to realize architecture evaluation without training and we adopt it to compute E-score. We simultaneously use N-score and E-score to obtain the initial population for the evolutionary algorithm. The specific calculation of the two scores is detailed in Appendix A.

**Score-based Selection.** As for N-score and E-score, we cannot pre-know the ranges nor the value distributions before computing them over a search space, thus it is difficult to directly combine them. Instead, We calculate these two scores to select BCs separately and then combine their results after removing duplicate BCs. In step one of BCNAS-SNN, we first randomly choose some candidate BCs to make up the initial population pool $P_{init}$. Then we leverage N-score and E-score to select some BCs with high scores from $P_{init}$, respectively, and combine them into the temporary population pool $P_{temp}$ as the initial population for network evaluation in the next step.

### 4.2.3 ACCURACY-BASED EVOLUTIONARY SEARCH .

In general, training of SNNs can be more time-consuming than that of ANNs due to the additional time dimension. Thus, an efficient search algorithm is crucial for discovering the optimal BCs in SNNs. In this study, we employ the evolutionary search algorithm as it has been shown to be capable of rapidly discovering architectures that are similar to optimal ones. In terms of architecture evaluation, zero-shot methods are known to have the lowest time consumption, but they are highly dependent on network initialization, which may cause inaccurate architecture evaluation. To mitigate this, we adopted the one-shot weight-sharing method, which measures candidate BCs by the validation accuracy of the BCSNN and allows for easy implementation due to the fixed ResNet backbone in our study.

The one-shot weight-sharing method in BCNAS-SNN is composed of two consecutive processes: training the super-network that contains all BCs, and searching for the optimal BCs according to the validation accuracy of the BCSNN obtained from the trained super-network. Given the trained super-network, BCNAS-SNN explores the search space by the evolutionary algorithm, throughout which there are two population pools, called the temporary population pool $P_{temp}$ and the top-$k$ population pool $P_{top}$, respectively. As described in subsubsection 4.2.2, we efficiently obtain the initialized $P_{temp}$ without training through the score-based selection. Then, we compute the validation accuracy of all architectures in $P_{temp}$ based on the trained super-network and choose $k$ architectures with the highest accuracy to make up the $P_{top}$. After these steps, $P_{temp}$ is updated by generating architectures through the evolutionary mechanism, including mutation and crossover.

**Mutation** A mutation at the placement of the BC is equivalent to a mutation at the positions of its ports. In order to discover the optimal BCs fast, we reduce the degree of freedom for mutations. We do not allow both ports of a BC to mutate at the same time, and the mutation position of each port is limited. Specifically, when a BC is mutated, only one port mutates to other closed ports with different probabilities, which is inversely proportional to the distance between them, and the position of the mutated port can not change by more than 2, as shown in Figure 4.

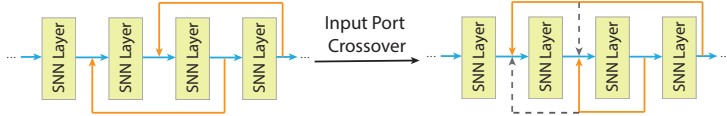

Figure 5: **Crossover in the evolutionary algorithm.** When the number of backward connections is two, the two backward connections swap their input ports to explore the search space.

Table 1: Comparisons with other exiting works on CIFAR10/100 datasets.

| Method | Use NAS | Architecture | Timestep1 | Accuracy1 | Timestep2 | Accuracy2 |
|---|---|---|---|---|---|---|
| **CIFAR10** | | | | | | |
| STBP-tdBN Zheng et al. (2021) | ✗ | ResNet-19 | 4 | 92.92 | 6 | 93.16 |
| Dspike Li et al. (2021) | ✗ | ResNet-18 | 4 | 93.66±0.05 | 6 | 94.25±0.07 |
| TET Deng et al. (2021) | ✗ | ResNet-19 | 4 | 94.44±0.08 | 6 | 94.50±0.07 |
| RecDis-SNN Guo et al. (2022) | ✗ | ResNet-19 | 4 | **95.53±0.05** | 6 | 95.55±0.05 |
| GLIF Yao et al. | ✗ | ResNet-19 | 4 | 94.85±0.07 | 6 | 95.03±0.08 |
| IM-Loss Guo et al. | ✗ | ResNet-19 | 4 | 95.40±0.08 | 6 | 95.49±0.05 |
| AutoSNN Na et al. (2022) | ✓ | AutoSNN(C=128) | - | - | 8 | 93.15 |
| SNASNet Kim et al. (2022) | ✓ | SNASNet-Bw | 5 | 93.73±0.32 | 8 | 94.12±0.25 |
| SpikeDHS Che et al. | ✓ | SpikeDHS-CLA(n3c5) | - | - | 6 | 95.50±0.03 |
| **BCNAS-SNN** | ✓ | ResNet-18 | 4 | **94.49±0.08** | 6 | **94.91±0.09** |
| **BCNAS-SNN** | ✓ | ResNet-19 | 4 | **95.27±0.08** | 6 | **95.67±0.04** |
| **CIFAR100** | | | | | | |
| Dspike Li et al. (2021) | ✗ | ResNet-18 | 4 | 73.35±0.14 | 6 | 74.24±0.10 |
| TET Deng et al. (2021) | ✗ | ResNet-19 | 4 | 74.47±0.15 | 6 | 74.72±0.28 |
| RecDis-SNN Guo et al. (2022) | ✗ | ResNet-19 | 4 | 74.10±0.13 | - | - |
| GLIF Yao et al. | ✗ | ResNet-19 | 4 | 77.05±0.14 | 6 | 77.35±0.07 |
| IM-Loss Guo et al. | ✗ | VGG-16 | 5 | 70.18±0.09 | - | - |
| AutoSNN Na et al. (2022) | ✓ | AutoSNN(C=64) | - | - | 8 | 69.16 |
| SNASNet Kim et al. (2022) | ✓ | SNASNet-Bw | 5 | 73.04±0.36 | - | - |
| SpikeDHS Che et al. | ✓ | SpikeDHS-CLA(n3s1) | - | - | 6 | 76.25±0.10 |
| **BCNAS-SNN** | ✓ | ResNet-18 | 4 | **74.85±0.12** | 6 | **75.48±0.24** |
| **BCNAS-SNN** | ✓ | ResNet-19 | 4 | **77.12±0.11** | 6 | **78.59±0.25** |
| **Tiny-Imagenet** | | | | | | |
| AutoSNN | ✓ | AutoSNN(C=64) | 8 | 46.79 | - | - |
| SNASNet | ✓ | SNASNet-Bw | 5 | 54.60±0.48 | - | - |
| **BCNAS-SNN** | ✓ | ResNet-18 | 5 | **63.43±0.07** | - | - |

**Crossover** There are two situations for crossover. When the number of backward connections is only one, crossover turns into addition, which randomly adds a new BC different from the existing one to the previous architecture. Otherwise, the two backward connections swap their input ports, as depicted in Figure 5.

For mutation and crossover, the parent architecture is randomly sampled from $P_{top}$ and the generated architectures are put into $P_{temp}$. When the number of architectures generated by mutation and crossover does not reach the size of $P_{temp}$, we use random sampling to supplement. The architectures in $P_{temp}$ are evaluated on the validation set. If the validation accuracy of the evaluated architectures is higher than those in $P_{top}$, $P_{top}$ is updated.

## 5 EXPERIMENTS

To demonstrate the effectiveness of our proposed BCNAS-SNN search framework, we conduct extensive experiments on CIFAR Krizhevsky et al. (2009), Tiny-ImageNet Deng et al. (2009) and CIFAR10DVS Li et al. (2017) with small time steps. Details regarding these datasets are provided in Appendix B. In subsection 5.2, we compare our results with other existing state of the art and in subsection 5.3, we carry out the ablation study to evaluate different aspects of our methods.

### 5.1 IMPLEMENTATION DETAILS

In the phase of super-network training, all training datasets are divided into 8:2 for $D_{train}$ and $D_{val}$. Adamw optimizer Loshchilov & Hutter (2018) is adopted for both super-network training and searched network training, of which the learning rate is 0.01 and the weight decay is 0.02. Meanwhile, we use cosine learning rate scheduling Loshchilov & Hutter (2016). We train all super-networks for

Table 2: Comparisons on CIFAR10DVS dataset. AutoSNN sets time steps as 20 and others use 10 time steps.

| Method | Architecture | Accuracy |
|---|---|---|
| STBP-tdBN Zheng et al. (2021) | ResNet-19 | 67.8 |
| Dspike Li et al. (2021) | ResNet-18 | 75.4±0.05 |
| TET Deng et al. (2021) | VGGSNN | **83.17±0.15** |
| TET Deng et al. (2021)[1] | ResNet-18 | 82.40±0.14 |
| RecDis-SNN Guo et al. (2022) | ResNet-19 | 72.42±0.06 |
| IM-Loss Guo et al. | ResNet-19 | 72.60±0.08 |
| AutoSNN Na et al. (2022) | AutoSNN(C=16) | 72.50 |
| **BCNAS-SNN** | ResNet-18 | **82.60±0.13** |

[1] Self-implementation results.

Table 3: Comparison between SNNs and BC-SNNs. The accuracy of BCSNNs is in the brackets.

| Dataset | Architecture | $T = 4$ | $T = 6$ |
|---|---|---|---|
| CIFAR10 | ResNet-18 | 94.01 (**94.58**) | 94.51 (**95.04**) |
| CIFAR10 | ResNet-19 | 94.88 (**95.18**) | 95.48 (**95.73**) |
| CIFAR10 | ResNet-18[1] | 71.39 (**73.33**) | 73.86 (**74.79**) |
| CIFAR100 | ResNet-18 | 73.38 (**74.97**) | 74.71 (**75.72**) |
| CIFAR100 | ResNet-19 | 75.24 (**77.18**) | 76.02 (**78.84**) |

[1] Self-implementation results.

100 epochs and train searched BCSNNs for 300, 300, 300, and 200 epochs on CIFAR10, CIFAR100, CIFAR100DVS, and Tiny-Imagenet, respectively.

As for the search algorithm, we set parameters as follows: the maximum number of search iterations $I$ is 20, the size of the initial pool $P_{init}$ is 2000, the size of the top-$k$ pool is 10, i.e., $k$ equals 10, the size of the temporal pool $P_{temp}$ is 20, the maximum number of architectures generated by mutation or crossover is 10, the mutation probability $\alpha$ is 0.2, and the crossover probability $\beta$ is 0.5.

## 5.2 COMPARISONS TO EXITING WORKS

**CIFAR10/100.** ResNet-18 He et al. (2016) and ResNet-19 Zheng et al. (2021) are adopted for these two datasets. As shown in Table 1, We report the mean and standard deviation of 3 runs under different random seeds. On CIFAR10, our searched BCSNN based on ResNet-18 achieves better results compared to Dspike and the best BCSNN based on ResNet-19 realizes the SOTA result when $T = 6$. Besides, the BCNAS-SNN search framework demonstrates a more excellent ability on CIFAR100. For both $T = 4$ and $T = 6$, the searched BCSNN based on ResNet-19 obtains SOTA results and has a 1.24% accuracy increment compared to GLIF, which achieved the SOTA result before when $T = 6$.

**Tiny-ImageNet.** Compared to CIFAR100, Tiny-ImageNet is more challenging with more training images belonging to 200 classes. In previous studies, there are two methods testing on this dataset, both of which use NAS methods to discover better architectures for SNNs. In our work, the searched BCSNN under ResNet-18 architecture outperforms the best of them by 8.83% accuracy, illustrating the superiority of the BCNAS-SNN search framework.

**CIFAR10DVS.** Different from CIFAR10/100 and Tiny-ImageNet, CIFAR10DVS is a neuromorphic dataset converted from CIFAR10, suffering much more noise than static datasets, which makes the well-trained SNN easy to overfit. Compared to the original SNN with TET loss, our searched BCSNN obtains slightly higher accuracy, which illustrates the appropriate backward connections can make better use of the time-domain information of SNNs. Deng et al. (2021) achieve the SOTA result and we think VGGSNN may be more suitable than ResNet for classification tasks on neuromorphic datasets such as CIFAR10DVS.

## 5.3 ABLATION STUDY

In this section, we demonstrate the effectiveness of our proposed BCNAS-SNN search framework through an ablation study, as shown in Table 4. We use ResNet-18 as the network backbone and search for BCs on the CIFAR100 dataset. All the experiments are conducted with 4 and 6 time steps.

**SNNs vs. BCSNNs.** We first compare the performance of searched BCSNNs with that of the primitive SNNs as shown in Table 3. The accuracy of BCSNNs is much better than the primitive SNNs. Especially on CIFAR100 dataset, we report 78.84% top-1 accuracy of ResNet-19 with optimal BCs, which is 2.82% better than the one without BCs. we conclude that the appropriate BCs can improve the SNNs' performance greatly and the results demonstrate the effectiveness of our search methods.

**N-Score & E-Score for selective initialization.** In step one of BCNAS-SNN, we leverage two metrics to select architectures from $P_{init}$ to make up initial $P_{temp}$. To verify the effectiveness of

Table 4: The results of our ablation study. Our method increases the accuracy of the SNN by $1.59\%$ and $1.01\%$ when $T = 4$ and $T = 6$, respectively, by adding appropriate BCs, which achieves the greatest improvement among all the experiments. The baseline refers to the experiment of the SNN without BCs. Exp is short for the experiment. The details of step 1 and step 2 are illustrated in Figure 3.

| Experiment | Method in Step1 | Method in Step2 | Search Way in Step2 | Size of $P_{init}$ | $T = 4$ | $T = 6$ |
|---|---|---|---|---|---|---|
| Baseline | - | - | - | - | 73.38 | 74.71 |
| **BCNAS-SNN** | E-score+N-score | One-Shot | Evolutionary | 2000 | **74.97** | **75.72** |
| Exp1 | N-score | One-Shot | Evolutionary | 2000 | 74.44 | 75.57 |
| Exp2 | E-score | One-Shot | Evolutionary | 2000 | 74.51 | 75.54 |
| Exp3 | E-score+N-score | One-Shot | Random | 2000 | 74.40 | 75.30 |
| Exp4 | E-score+N-score | N-score | Evolutionary | 2000 | 73.56 | 74.75 |
| Exp5 | E-score+N-score | E-score | Evolutionary | 2000 | 73.94 | 74.98 |
| Exp6 | E-score+N-score | One-Shot | Evolutionary | 1000 | 74.26 | 75.27 |
| Exp7 | E-score+N-score | One-Shot | Evolutionary | 500 | 73.51 | 74.97 |
| Exp8 | E-score+N-score | One-Shot | Evolutionary | 0 | 72.58 | 74.72 |

the proposed method, we make initial $P_{temp}$ composed of architectures with high scores calculated only by N-score or E-score. From Exp1 and Exp2 in Table 4, we can learn that all of these methods obtain good performance, but the combined method gets a higher accuracy. It demonstrates that the combined zero-shot method is less fortuitous than the single zero-shot method.

**Evolutionary Search vs. Random Search.** To verify the effectiveness of the evolutionary algorithm, we keep the initial $P_{temp}$ and use random search instead of the evolutionary search in the step two of BCNAS-SNN. From Exp3 in Table 4, Obviously, it is more efficient to adopt the evolutionary algorithm than random search. Meanwhile, the results of these experiments illustrate our designed evolutionary search algorithm is effective.

**One-Shot vs. Zero-Shot.** In this work, considering the time consumption, we adopt zero-shot methods in step one of BCNAS-SNN. Here, the one-shot weight-sharing method and zero-shot methods are discussed within step two of BCNAS-SNN. We test how the results change when adopting zero-shot methods in the evolutionary process. Exp4 and Exp5 in Table 4 show that the one-shot weight-sharing method obtains better results and the accuracy gap between it and zero-shot methods is obvious. Through these experiments, we conclude that it is difficult to find an optimal BCSNN only according to zero-shot methods, although they can save time a lot. And the adoption of the one-shot weight-sharing method is necessary.

**Effect of Search Budget.** We investigate the effect of the search budget, which refers to the number of evaluated architectures throughout the search process. For simplicity, we fix the maximum number of search iterations $I$ and accomplish some experiments with different sizes of $P_{init}$. From Exp6 to Exp8 in Table 4, we can conclude that the accuracy of the final searched BCSNNs will decrease as the number of samples decreases. In particular, when the size of $P_{init}$ is reduced to 0, i.e., step one of BCNAS-SNN is removed and initial $P_{temp}$ is randomly sampled, the accuracy drops a lot, even lower than the baseline. Thus in our work, keeping enough samples in $P_{init}$ is the key to final network performance.

## 6 CONCLUSION

This paper explores the potential benefits of incorporating BCs in SNNs, which can potentially improve performance by providing extra-temporal information. We find that not all BCs are effective and develop the BCNAS-SNN search framework to identify optimal BCs through a two-step search strategy. Our searched BCSNNs have yielded state-of-the-art results on the CIFAR10/100 and Tiny-ImageNet datasets. Interestingly, we find the searched BCs prefer to be connected to the first two layers, which may motivate future architecture design of SNNs. Extensive experiments demonstrate the effectiveness and efficiency of our proposed method. Currently, considering energy consumption and time cost, we apply the fixed operations and limit the number of BCs, and a larger search space remains to be discussed and discovered in future work.

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

## A  THE CALCULATION OF N-SCORE AND E-SCORE

**N-score.** NASWOT Mellor et al. (2021) observes that the architecture having distinctive representations across different data samples is likely to achieve higher performance. The method is based on the theory of linear regions Montúfar (2017); Hanin & Rolnick (2019) for ReLU networks. We apply this approach to SNNs based on Spike-Aware Hamming Distance(SAHD) Kim et al. (2022) to compute N-score in this work. We define a binary indicator according to the value of the membrane potential. If the value is greater than the threshold, the LIF neuron is mapped to 1; otherwise 0. A layer in SNNs is encoded to a binary vector $\mathbf{c}$ based on the mapping results. We obtain the accurate distance $d(\mathbf{c}_i, \mathbf{c}_j)$ between different samples $i$ and $j$ by SAHD. Given $N$ samples in one mini-batch, we construct a kernel matrix at each time step as follows:

$$\mathbf{K}_t = \begin{pmatrix} N_A - d(\mathbf{c}_1, \mathbf{c}_1) & \cdots & N_A - d(\mathbf{c}_1, \mathbf{c}_N) \\ \vdots & \ddots & \vdots \\ N_A - d(\mathbf{c}_N, \mathbf{c}_1) & \cdots & N_A - d(\mathbf{c}_N, \mathbf{c}_N) \end{pmatrix} \tag{7}$$

Here, $N_A$ represents the number of LIF neurons in the given layer. The final score of the candidate architecture is:

$$s = \log(\det|\sum_t \sum_l \mathbf{K}_t^l|), \tag{8}$$

where, $l$ stands for the layer index.

**E-score.** Focusing on the intra-class and inter-class correlations of an initialized network, EPE-NAS Lopes et al. (2021) realizes architecture evaluation without training. It evaluates how the gradients behave with respect to the input to eliminate the need for training, and we use it to calculate E-score in this work. The linear map is defined as:

$$\mathbf{w}_i = \frac{\partial f(\mathbf{x}_i)}{\partial \mathbf{x}_i}, \tag{9}$$

where $\mathbf{x}_i$ represents the input and $f(\mathbf{x}_i)$ is the network. The Jacobi matrix for different data points can be calculated as:

$$\mathbf{J} = \left( \frac{\partial f(\mathbf{x}_1)}{\partial \mathbf{x}_1}, \quad \cdots, \quad \frac{\partial f(\mathbf{x}_N)}{\partial \mathbf{x}_N} \right)^\top, \tag{10}$$

where $N$ denotes the number of data samples. A network with good expressiveness should simultaneously be able to distinguish local linear operators for each data point and have similar results for similar data points. To evaluate this behavior, we compute a covariance matrix for each class:

$$\mathbf{C}_{J_c} = (\mathbf{J} - \mathbf{M}_{J_c})(\mathbf{J} - \mathbf{M}_{J_c})^\top, \tag{11}$$

where $\mathbf{M}_{J_c}$ is the matrix with elements:

$$(\mathbf{M}_{J_c})_{i,j} = \frac{1}{N} \sum_{\substack{n \in \{1, \cdots, N\} \\ x_i \in class\, c}} \mathbf{J}_{i,n}, \tag{12}$$

where $c$ stands for the class, $c \in \{1, \cdots, C\}$, and $C$ is the number of classes in the given batch. Then, we calculate the correlation matrix $\mathbf{\Sigma}_{J_c}$ per class as follows:

$$(\Sigma_{J_c})_{i,j} = \frac{(\mathbf{C}_{J_c})_{i,j}}{\sqrt{(\mathbf{C}_{J_c})_{i,i}(\mathbf{C}_{J_c})_{j,j}}}. \tag{13}$$

These correlation matrices may have different sizes due to the different number of data points in each class, so we make some adjustments as follows:

$$\mathbf{E}_c = \begin{cases} \sum_{i=0}^N \sum_{j=0}^N \log(|(\Sigma_{J_c})_{i,j}| + k), & \text{if } C \le 100 \\ \frac{\sum_{i=0}^N \sum_{j=0}^N \log(|(\Sigma_{J_c})_{i,j}| + k)}{||\Sigma_{J_c}||}, & \text{otherwise} \end{cases} \tag{14}$$

where $k$ is a small constant, the value of which is $1 \times 10^{-5}$. $|\cdot|$ denotes the number of elements of a set. The final score can be formulated by:

$$s = \begin{cases} \sum_{i=0}^C |\mathbf{E}_i|, & \text{if } C \le 100 \\ \frac{\sum_{i=0}^C \sum_{j=i}^C |\mathbf{E}_i - \mathbf{E}_j|}{||\mathbf{E}||}, & \text{otherwise} \end{cases} \tag{15}$$

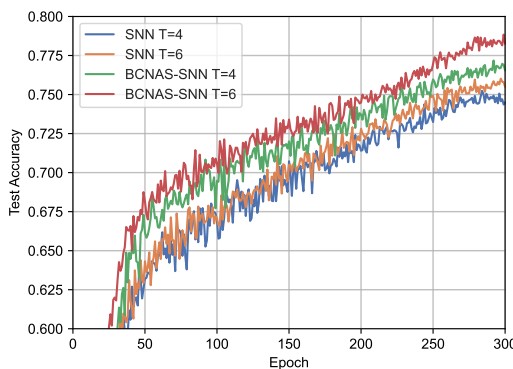

Figure 6: Test accuracy vs. training epochs for vanilla SNNs and BCNAS-SNNs.

## B  DATASETS

CIFAR10 and CIFAR100 include images with a resolution of 32×32 and 3 channels, in which the images are divided into 10 categories and 20 categories. They are composed of 50,000 training data and 10,000 test data. Tiny-ImageNet includes images with a resolution of 64×64 and 3 channels, where the images are sampled from ImageNet Russakovsky et al. (2015) and downsized from 224x224 to 64×64. 100K images, 2.5K, and 2.5K images are used for training, validating, and testing, respectively. Since the data used for testing is not labeled, we use the data for validating as our test data. For these static datasets, we use common data processing methods, including normalization, the central padding of images to 40×40 and then random cropping back to 32×32, and random horizontal flipping. CIFAR10DVS is a neuromorphic dataset, which includes data with a format of the event stream. It is collected by a dynamic vision sensor (DVS), which outputs 128×128 images with 2 channels. 10,000 images from CIFAR10 are converted into the spike trains, and there are 1,000 images per class. We divide the dataset into training data and test data by 9:1 and resize each image to 48×48. We use the same data processing method as Deng et al. (2021).

### B.1  CONVERGENCE

In this section, we visualize the convergence curve of our BCNAS-SNNs and the vanilla SNNs. We test ResNet-19 on the CIFAR-100 dataset with 4 or 6 time steps. Figure 6 shows the convergence speed difference between our method and the vanilla SNN. Specifically, BCNAS-SNN with 4 time steps can have a better convergence than a 6-time-step vanilla SNN, demonstrating the effectiveness of our method.

## C  ALL SEARCHED BACKWARD CONNECTIONS BASED ON RESNETS

In our work, we search for optimal backward connections based on ResNets across different datasets with 4 and 6 time steps. ResNets include ResNet-18, ResNet-19, and ResNet-18 without residual connections. The datasets include CIFAR10, CIFAR100, Tiny-ImageNet, and CIFAR10DVS. All searched results are shown in Figure 7 and Figure 8. An intriguing phenomenon can be easily discovered the searched backward connections prefer to be connected to the beginning of the SNN.

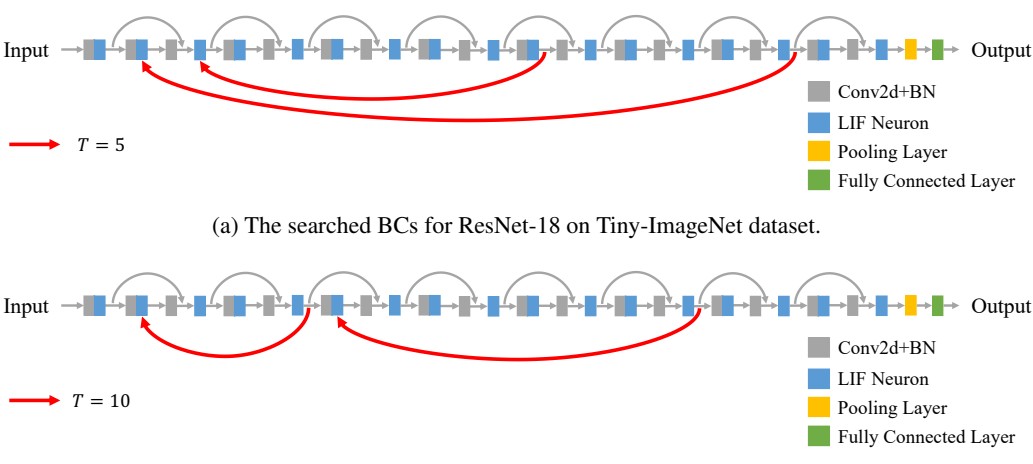

(a) The searched BCs for ResNet-18 on Tiny-ImageNet dataset.

(b) The searched BCs for ResNet-18 on CIFAR10DVS dataset.

Figure 7: The searched BCs for ResNet-18 on Tiny-ImageNet and CIFAR10DVS datasets.

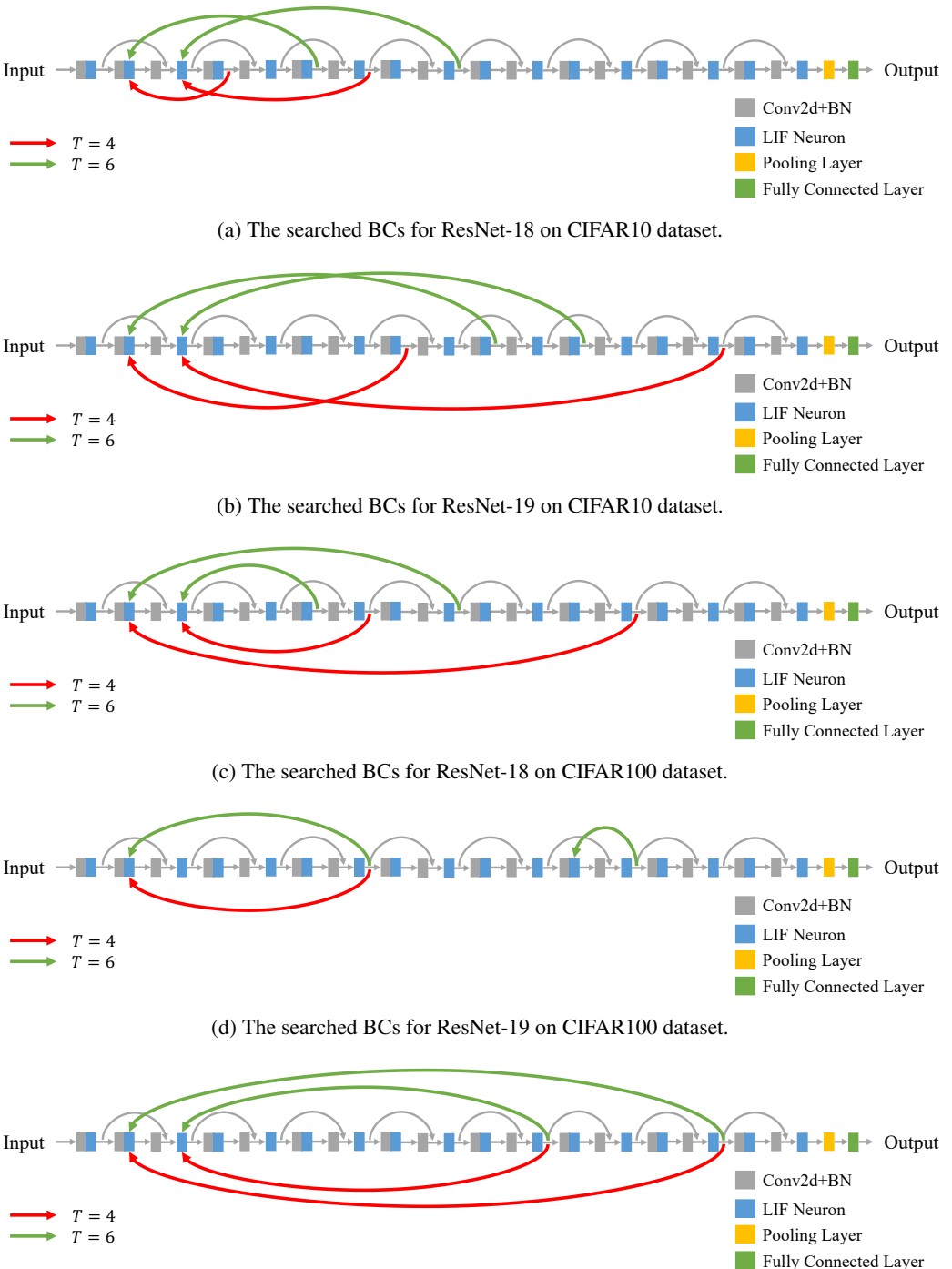

(a) The searched BCs for ResNet-18 on CIFAR10 dataset.

(b) The searched BCs for ResNet-19 on CIFAR10 dataset.

(c) The searched BCs for ResNet-18 on CIFAR100 dataset.

(d) The searched BCs for ResNet-19 on CIFAR100 dataset.

(e) The searched BCs for ResNet-18 without residual connections on CIFAR100 dataset.

Figure 8: The searched BCs for ResNets on CIFAR10/100 datasets.

