# OpenReview forum: "Proper Backward Connection Placement Boosts Spiking Neural Networks"
_ICLR.cc/2024/Conference — Submitted to ICLR 2024_

### Official Review · Reviewer_6HHN · 2023-10-13

**Soundness:** 3 good
**Presentation:** 4 excellent
**Contribution:** 2 fair
**Rating:** 5
**Confidence:** 4

**Summary:**

This paper proposes a framework called BCNAS-SNN that automatically searches for the optimal placement of global backward connections (BCs) in spiking neural networks (SNNs). They develop the search space and implement an effective BC search using an evolutionary algorithm and a two-step strategy. The authors show that BCNAS-SNN can discover effective BCs that improve the performance of SNNs on various datasets, such as CIFAR10, CIFAR100, Tiny-ImageNet, and CIFAR10DVS.

**Strengths:**

The paper proposes a novel method to search for optimal backward connections in spiking neural networks. The paper introduces a two-step search strategy to efficiently explore the large search space of BCs placement. The paper also reveals an interesting phenomenon that the searched BCs prefer to be connected to the front layers of SNNs, which may inspire future architecture design of SNNs.

The paper is well-written and organized, with clear definitions and notations. The paper uses figures and tables to illustrate the concept of BCs and the results of the experiments.

**Weaknesses:**

We noticed that the performance increment is limited compared to previous works [1,2]. On dataset CIFAR-100, Guo et al [1] achieved 79.51% accuracy while this work reported a 78.59% average accuracy. Also, on dataset CIFAR10-DVS, this work reported an 82.60% accuracy, which is only 0.2% higher than Deng et al [2]. Also, although costly, it would be more convincing if authors post the result on ImageNet dataset, which is a more popular and large-scale dataset.

[1] Guo, Y., Zhang, Y., Chen, Y., Peng, W., Liu, X., Zhang, L., ... & Ma, Z. (2023). Membrane Potential Batch Normalization for Spiking Neural Networks. ICCV.

[2] Deng, S., Li, Y., Zhang, S., & Gu, S. (2022). Temporal Efficient Training of Spiking Neural Network via Gradient Re-weighting. ICLR.

Authors should demonstrate the extra training cost of the architecture search. In my understanding, the NAS usually requires more training cost. However, the performance improvement of this work is subtle. Whether using the extra training resources is worth the small performance gain?

Authors should explain why they choose to perform architecture search on backward connections and why adding BCs can improve the performance. What is the advantage of BC search compared with other NAS methods?

**Questions:**

The results show that the searched BCs prefer to be connected to the front layers of SNNs. Is it because adding such connection can alleviate the problem of gradient explosion or gradient disappearance in deep networks?

---

### Official Review · Reviewer_FSBw · 2023-10-24

**Soundness:** 2 fair
**Presentation:** 2 fair
**Contribution:** 2 fair
**Rating:** 3
**Confidence:** 5

**Summary:**

This study explores the potential benefits of introducing backward connections (BCs) in Spiking Neural Networks (SNNs) and proposes the BCNAS-SNN search framework to automatically identify the optimal BCs. The research findings indicate that BCSNN achieves state-of-the-art results on the CIFAR10/100 and Tiny-ImageNet datasets, and it is observed that BCs tend to be connected to the first two layers.

**Strengths:**

The research topic is forward-looking, exploring the potential benefits of BCs in SNNs. It introduces the BCNAS-SNN search framework for automatically searching for the optimal BCs.

**Weaknesses:**

There is a missing part on the right side in Figure 1. The validation of the experimental section is not sufficiently thorough. Additional experimental validation regarding non-residual structures such as VGG can be included. The method's application in ANN and SNN can be compared. A comparison with the accuracy without backward connections can be added in Table 1.

**Questions:**

Is this method constrained by computational resources, or are there plans to further optimize it for improved efficiency?

---

### Official Review · Reviewer_SWJe · 2023-10-31

**Soundness:** 3 good
**Presentation:** 3 good
**Contribution:** 3 good
**Rating:** 5
**Confidence:** 4

**Summary:**

This study investigates the impact of backward connections (BCs) on Spiking Neural Networks (SNNs) and introduces the Backward Connection Neural Architecture Search (BCNAS-SNN) framework to optimize BC placement. The research contributes by examining the effects of global backward connections in SNNs and proposing a novel search space for BC-based SNNs. Ablation studies are conducted to analyze design components.

**Strengths:**

1. This study is the first to comprehensively investigate the varying effects of global backward connections in SNNs.
2. It introduces a two-step search strategy to reduce the search space of BCs.

**Weaknesses:**

Please refer to `Questions`.

**Questions:**

1. I would like to ascertain the precise definition of "BCSNN." Is it an abbreviation for "Backward Connection SNN" and does it refer to subnets extracted from a supernet?


2. The research lacks an analysis of time series data. While backward connections may exhibit potential advantages in time series data, their performance in static data is unclear. Some studies suggest that BCs perform similarly to forward connections in static data but offer improved biological relevance. Has the author conducted experiments on time series data like DVS gesture recognition to address this aspect, instead of focusing solely on static data?

3. The analysis of the search process is not comprehensive.
   3.1. Is it established that 100 epochs of training are sufficient to fully converge the supernet? Has any experimentation been done to establish the relationship between the performance of a subnet extracted from the trained supernet and a subnet trained from scratch until full convergence? Is there a strong positive correlation between the two?
   3.2. How many different seed values were employed in the ablation study? The experiment showed a slight improvement in performance, so it's important to discern whether this improvement can be attributed to the variation in random seed values or if it is linked to the different search methods applied.
   3.3. In Experiment 8 of the ablation study, it would be valuable to provide a comparison by conducting the same experiment but with a random search strategy in step 2, thereby comparing evolutionary search to random search.
   3.4. Given that the evolutionary search in Experiment 8 appears less effective, could the author include a plot depicting the search iteration versus accuracy to help illustrate the search process?

---

### Meta-Review · Area_Chair_Bska · 2023-12-10

**Metareview:**

I'm very sorry that this paper could not be accepted to ICLR this year. Although the ideas seem promising and reviewers appreciated many strengths in the manuscript (eg, clearly written, original, the first to the effects of global backward connections in SNNs), they ultimately did not judge it to be above the bar for acceptance to this year's meeting. I wish the authors the best of luck in revising this work for publication elsewhere.

**Justification For Why Not Higher Score:**

The reviewers were unanimous that it should be rejected, and authors didn't write rebuttals.

**Justification For Why Not Lower Score:**

Can't be lower than this.

---

### Decision · Program_Chairs · 2024-01-16

Reject